# Visual imitation with a minimal adversary

## Abstract

High-dimensional sparse reward tasks present major challenges for reinforcement learning agents. In this work we use imitation learning to address two of these challenges: how to learn a useful representation of the world e.g. from pixels, and how to explore efficiently given the rarity of a reward signal? We show that adversarial imitation can work well even in a high dimensional observation space. Surprisingly, the discriminator network, providing a learned reward function, can be tiny, comprising as few as 128 parameters, and can be easily trained using the most basic GAN formulation. Our approach removes limitations present in most contemporary imitation approaches: requiring no demonstrator actions (only video), no special initial conditions or warm starts, and no explicit tracking of any single demo. The proposed agent can solve a challenging robot manipulation task of block stacking from only video demonstrations and sparse reward, in which the non-imitating agents fail to learn completely. Furthermore, our agent learns much faster than competing approaches that depend on hand-crafted, staged dense reward functions, and also better compared to standard GAIL baselines. Finally, we develop a new adversarial goal recognizer that in some cases allows the agent to learn stacking without any task reward, purely from imitation.

## 1 Introduction

Many real world tasks, especially in robotics, involve high-dimensional inputs such as video pixels, and may have sparse rewards or even no rewards. This is a challenging problem for training reinforcement learning agents, because a randomly-initialized agent may need to interact with the environment for a very long time before receiving any learning signal. On a physical robot, this problem is compounded by the small number of actors gathering experience (typically 1), the slow speed compared to simulation, and the high cost of an episode that goes wrong.

One way to make this exploration problem easier is to create a dense reward function. Often this is done by human hand-crafting of a staged reward. For example, in a block stacking task, the rewards can be provided in increasing magnitude for a robotic arm reaching toward a block, then more reward for grasping and lifting, then more for placing one on top of the other, and finally more for removing the arm from the blocks. However, this requires human expertise to design, and also requires access to the state of objects in the environment. Further, it can lead to sub-optimal policies. In the case of learning from pixels only, these stages may also be difficult to distinguish perfectly.

Instead of hand-crafting a dense reward function, another popular approach is to collect expert demonstrations, and use these to derive a dense reward function, e.g. via imitation. Contemporary imitation learning approaches have a variety of trade-offs and requirements. For example, many approaches require expert actions in addition to the observations, or access to the expert states in order to perform special initialization, tracking, or warm starting (Hosu & Rebedea (2016)) the agent from the later stages of expert trajectories.

Generative Adversarial Imitation Learning (GAIL, Ho & Ermon (2016)), one of the most popular reinforcement learning methods for imitation, does not have these particular limitations. In GAIL, a discriminator network learns to distinguish agent from expert trajectories; the agent is then rewarded for "fooling" this discriminator into thinking its behavior is that of the expert. However, GAIL tends to be inefficient in terms of environment interactions, is typically trained on-policy, and to our knowledge has not been shown to solve challenging high-dimensional sparse reward tasks. This might be because intuitively, in high dimensions such as pixel space, there are simply too many

minute differences between the agent and expert for a deep neural network discriminator to exploit. Furthermore, optimizing deep adversarial networks is notoriously difficult, although there is progress towards stabilizing this problem; see e.g. Arjovsky & Bottou (2017); Arjovsky et al. (2017).

In this work, we show that if GAIL is provided the right kind of features, it can actually easily handle high-dimensional pixel observations using a *single layer* discriminator network. We can also improve its efficiency in terms of environment interactions by using a Deep Distributed Deterministic Policy Gradients (D4PG) agent (Barth-Maron et al., 2018), which is a state-of-the-art off-policy method for control, that can take advantage of a replay buffer to store past experiences.

We show that several types of features can be used successfully with a tiny, single-layer adversary, in cases where a deep adversary on pixels would fail completely. Specifically: self-supervised embeddings, e.g. Contrastive Predictive Coding (van den Oord et al., 2018); surprisingly, random projections through a deep residual network; and value network features from the D4PG agent itself. Concurrently with (Kostrikov et al., 2018), we additionally show how to modify GAIL for the off-policy case of D4PG agents with experience replay.

In our experiments, we demonstrate that our proposed approach is able to, from pixels, solve a challenging simulated robotic block stacking task using only demonstrations and a sparse binary reward indicating whether or not the stack has completed. Previous imitation approaches to this task have all used dense, staged task rewards crafted by humans and/or true state instead of pixels. In addition to reducing the dependency on hand-crafted rewards, our approach learns to stack faster than the dense staged reward baseline agent with the same amount of actor processes.

The main contributions of this paper are the following:

- A 6-DoF Jaco robot arm agent that learns block stacking from demonstration videos and sparse reward. On a simulated Jaco arm it achieves $94\%$ success rate, compared to $\sim 29\%$ using a behavior cloning agent with an equivalent number of demonstrations.

- An adversary-based early termination method for actor processes, that improves task performance and learning speed by creating a natural curriculum for the agent.

- An agent that learns with no task reward using an auxiliary goal recognizer adversary, achieving $55\%$ stacking success from video imitation only.

- Ablation experiments on Jaco stacking as well as a 2D planar walker benchmark (Tassa et al., 2018), to understand the specific reasons for improvement in our agent. We find that random projections with a linear discriminator work surprisingly well in some cases, and that using value network features is even better.

## 2 BACKGROUND

Following the notation of Sutton et al. (1998), a Markov Decision Process (MDP) is a tuple $(\mathcal{S}, \mathcal{A}, R, P, \gamma)$ with states $\mathcal{S}$, actions $\mathcal{A}$, reward function $R(s, a)$, transition distribution $P(s'|s, a)$, and discount $\gamma$. An agent in state $s \in S$ takes action $a \in A$ according to its policy $\pi$ and moves to state $s' \in \mathcal{S}$ according to the transition distribution. The goal of training an agent is to find a policy that maximizes the expected sum of discounted rewards, represented by the action value function $Q^\pi(s, a) = \mathbb{E}^\pi[\sum_{t=0}^\infty \gamma^t R(s_t, a_t)]$, where $\mathbb{E}^\pi$ is an expectation over trajectories starting from $s_0 = s$ and taking action $a_0 = a$ and thereafter running the policy $\pi$.

### 2.1 DEEP DETERMINISTIC POLICY GRADIENTS (DDPG)

DDPG (Lillicrap et al., 2016) is an actor-critic method in which the actor, or policy $\pi(s|\phi)$ and the critic, or action-value function $Q(s, a|\theta)$ are represented by neural networks with parameters $\phi$ and $\theta$, respectively. New transitions $(s, a, r, s')$ are added to a replay buffer $\mathcal{B}$ by sampling from the policy according to $a = \pi(s|\phi) + \epsilon$, with $\epsilon \sim \mathcal{N}$ added for better exploration.

The action-value function is trained to match the 1-step returns by minimizing

$$\mathcal{L}_1 = \mathbb{E}_{(s,a,r,s') \sim \mathcal{B}}[R(s, a) + \gamma Q'(s', \pi'(s'|\phi')|\theta') - Q(s, a|\theta)]^2 \tag{1}$$

Figure 1: Actor and learner process pseudocode.

---

**Algorithm 1** Actor process

Replay buffer $\mathcal{B}$, environment $\mathcal{E}$
**for** $n_{episodes}$ **do**
  **for** $t = 0$ **to** $T$ **do**
    $a_t \leftarrow \pi(s_t)$  $r_t, s_{t+1} \leftarrow \mathcal{E}(s_t, a_t)$
    $\mathcal{B} \leftarrow (s_t, a_t, r_t, s_{t+1})$ // Add to replay
    $r_{imitation} \leftarrow D(s_{t+1})$
    **if** $r_{imitation} < \beta$ **break**
  **end for**
**end for**

---

**Algorithm 2** Learner process

Replay buffer $\mathcal{B}$, expert dataset $\mathcal{D}$
**for** $n_{updates}$ **do**
  // Sample from replay and expert data.
  $(s, a, r, s') \sim \mathcal{B}$,  $s_{expert} \sim \mathcal{D}$
  // Add imitation to reward.
  $r = \alpha r + (1 - \alpha)D(s')$
  // Optimize Equations 1, 2, and 4.
  $Q, \pi, D \leftarrow Update(s, a, r, s', s_{expert})$
**end for**

---

where the transition is sampled from the replay buffer $\mathcal{B}$, and $\pi', Q'$ are target actor and critic networks parameterized by $\phi'$ and $\theta'$ respectively. To improve the stability of weight updates, the target networks are updated every $K$ learning steps.

The policy network is trained via gradient descent to produce actions that maximize the action-value function using the deterministic policy gradient (Silver et al., 2014):

$$\nabla_\phi J(\phi) \approx \mathbb{E}_{(s,a) \sim \mathcal{B}}[\nabla_\phi Q(s, \pi(s|\phi))] \tag{2}$$

Building on top of the basic DDPG agent, we also leverage several subsequent improvements following Barth-Maron et al. (2018), called D4PG. We summarize these in section 7.6.

## 2.2 GENERATIVE ADVERSARIAL IMITATION LEARNING (GAIL)

In GAIL, a reward function is learned by training a discriminator network $D(s, a)$ to distinguish between agent transitions and expert transitions. The GAIL objective is formulated as follows:

$$\max_\pi \min_D \mathbb{E}_\pi[\log D(s, a)] + \mathbb{E}_{\pi_E}[\log(1 - D(s, a))] + \lambda H(\pi) \tag{3}$$

where $\pi$ is the agent policy, $\pi_E$ the expert policy, and $H(\pi)$ an entropy regularizer. GAIL is closely related to MaxEnt inverse reinforcement learning (Ziebart et al., 2008; Finn et al., 2016).

## 3 MODEL

To make use of all the available training data, we use a D4PG agent that does off-policy training with experience replay with buffer $\mathcal{B}$. The actor and critic updates are the same as in D4PG, but in addition we jointly train the reward function using a modified equation 3 for $D$:

$$\min_D \mathbb{E}_{s \sim \mathcal{B}}[\log D(s)] + \mathbb{E}_{\pi_E}[\log(1 - D(s))] \tag{4}$$

As pointed out by Kostrikov et al. (2018), the use of a replay buffer changes the original expectation over the agent policy in equation 3, so that the discriminator must distinguish expert transitions from the transitions produced by all previous agents. This could be corrected by importance sampling, but in practice, we also find this not to be necessary.

Note that we do not use actions $a$ in the discriminator because we only assume access to videos of the expert. Our reward function interpolates imitation reward and a sparse task reward:

$$R(s, a, s') = \alpha R_{task}(s, a, s') + (1 - \alpha)D(s') \tag{5}$$

where $s'$ is the state reached after taking action $a$ in state $s$.

Note that by $D(.)$ we mean the sigmoid of the logits, $\sigma(x) = 1/(1 + \exp(-x))$, which bounds the imitation reward between 0 and 1. This is convenient because it allows us to choose intuitive values for early termination of episodes in the actor process, e.g. if the discriminator score is too low, and it is scaled similarly to the sparse reward that we use in our block stacking experiments.

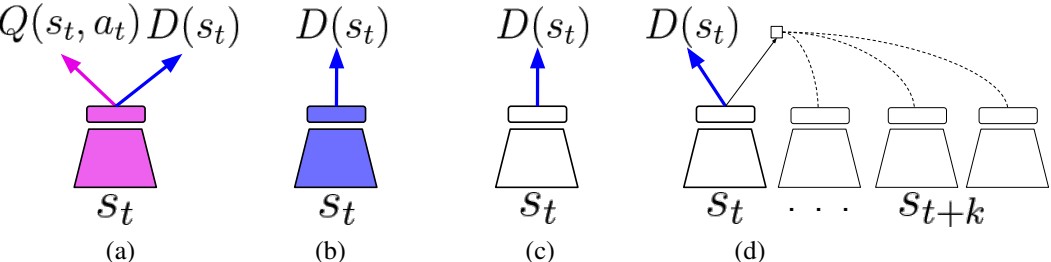

Figure 2: Features used as input to the reward function. Blue color indicates the flow of gradient from the discriminator objective. In all cases, $D(s)$ is a single layer network projecting the features onto a scalar score. (a) Using features from the value network, which are also changing online with the discriminator. (b) Using pixels directly; i.e. $D(s)$ becomes a deep convolutional network. We use the same architecture as for the value network. (c) Random projections - same as (b) except that gradients do not flow into the convolutional network. (d) Using features learned by contrastive predictive coding on the expert demonstrations. See section 3.2 for details about the pre-training.

Above we include pseudocode for the actor and learner processes. In practice we use many CPU actor processes in parallel (128 in our experiments) and a single learner process on GPU. The actor processes receive updated network parameters every 100 acting steps.

An important detail in the actor is the use of early termination of the episode when the discriminator score is below a threshold $\beta$, which is a hyperparameter. This prevents the agent from drifting too far from the expert trajectories, and avoids wasting computation time. In practice we set $\beta = 0.1$. We include a plot of the training dynamics induced by this early stopping mechanism in figure 7b.

## 3.1    REWARD FUNCTION NETWORK ARCHITECTURE

A critical design choice in our agent is the type of network used in the discriminator, whose output is used as the reward function. If the network has too much capacity and direct access to high-dimensional observations, it may be too easy for it to distinguish agent from expert, and too difficult for the agent to fool the discriminator. If the discriminator has too little capacity, it may not capture the salient differences between agent and expert, and so cannot teach the agent how to solve the task. Figure 2 illustrates the discriminator architectures that we study in this work.

## 3.2    PRE-TRAINING FEATURES FROM EXPERT DEMONSTRATIONS

Expert demonstrations are a useful source of data for feature learning, because by construction they have a sufficient coverage of regions of state space that the agent needs to observe in order to solve the task, at least for the problem instances in the training set. We do not assume access to expert actions, so behavior cloning is not an option for feature learning. Also, we assume that the images are high resolution relative to what contemporary generative models are capable of generating realistically and efficiently, so we decided *not* to learn features by predicting in pixel space. Furthermore, pixel prediction objectives may not encourage the learning of long term structure in the data, which we expect to be most helpful for imitation learning.

Based on these desiderata, contrastive predictive coding (CPC, van den Oord et al. (2018)) is an appealing option. CPC is a representation learning technique that maps a sequence of observations into a latent space such that a simple autoregressive model can easily make long-term predictions over latents. Crucially CPC uses a probabilistic contrastive loss using negative sampling. This allows the encoder and autoregressive model to be trained jointly and without having to need a decoder model to make predictions at the observation level. We describe CPC in more detail in section 7.3

## 3.3    LEARNING WITH NO TASK REWARD

Beyond moving from hand-crafted dense staged rewards to sparse rewards, we can also remove task rewards entirely. One straightforward way to achieve this could be to swap out the "success or failure" sparse reward with a neural network goal recognizer, trained on expert trajectories. The problem here

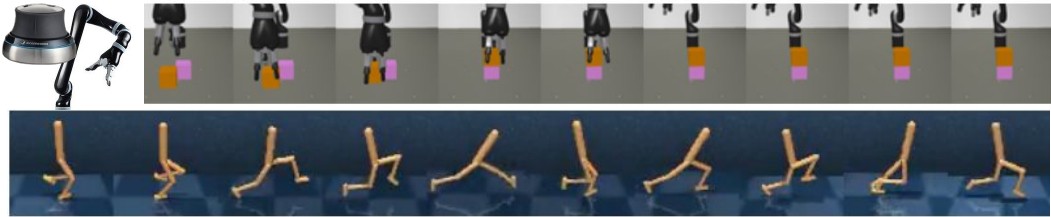

Figure 3: Environments for visual imitation experiments. Top row: simulated Jaco arm with SpaceNavigator teloperation. Bottom row: planar walker running at a target velocity.

is, such a network would be frozen during agent training, and the agent could adversarially find blind spots of the goal recognizer, and get imitation reward without solving this task. In fact, this is what we observe in practice (see failure cases in the appendix).

To overcome this issue, we can replace the sparse task reward with another discriminator, whose job is to detect whether an agent has reached a goal state or not. A goal state can be defined for our purposes as a state in the latter $1/M$ proportion of the expert demonstration. In our experiments we used $M = 3$. The modified reward function then becomes

$$R(s, a, s') = \alpha D_{goal}(s') + (1 - \alpha)D(s') \tag{6}$$

where $D_{goal}$ is the secondary goal discriminator network. It does not share weights with $D$, but it is also a single-layer network that operates on the same feature space as $D$. Training $D_{goal}$ is the same as for $D$, except that the expert states are only sampled from the latter $1/M$ portion of each demonstration trajectory.

Typically GAIL is viewed as purely an imitation learning method, which bounds the agent performance by how well the demonstrator performs. However, by training a second discriminator to recognize whether a goal state has been reached, it is possible for an agent to surpass the demonstrator by learning to reach the goal faster, which has already observed when agents are trained with combined imitation and sparse task rewards.

## 4 EXPERIMENTS

Our environments are visualized in figure 3. The first consists of a Kinova Jaco arm, and two blocks on a tabletop. The arm has 9 degrees of freedom: six arm joints and three actuated fingers. Policies control the arm by setting the joint velocity commands, producing 9-dimensional continuous velocities in the range of [-1, 1] at 20Hz. The observations are 128x128 RGB images. The hand-crafted reward functions (sparse and dense staged) are described in section 7.4.

To collect demonstrations we use a SpaceNavigator 3D motion controller. A human operator controlled the jaco arm with a position controller, and gathers 500 episodes of demonstration for each task including observations, actions, and physical states. Another 500 trajectories (used for validation purposes) were gathered by a different human demonstrator. A dataset of 30 "non-expert" trajectories (used for CPC diagnostics) were collected by performing behaviors other than stacking, such as random arm motions, lifting and dropping blocks, and stacking in an incorrect orientation.

The second environment is a 2D walker from the DeepMind control suite (Tassa et al., 2018). To collect demonstrations, we trained a D4PG agent from proprioceptive states to match a target velocity. Our agents use $64 \times 64$ pixel observations of 200 expert demonstrations of length 300 steps.

For all of the D4PG agents we used the hyperparameter settings listed in section 7.5.

### 4.1 JACO ARM BLOCK STACKING WITH SPARSE REWARD

In this section we compare our imitation method to a comparable D4PG agent on dense and sparse reward, and to comparable GAIL agents with discriminator networks operating on pixels directly. Figure 4 shows that our proposed method using a tiny adversary compares favorably.

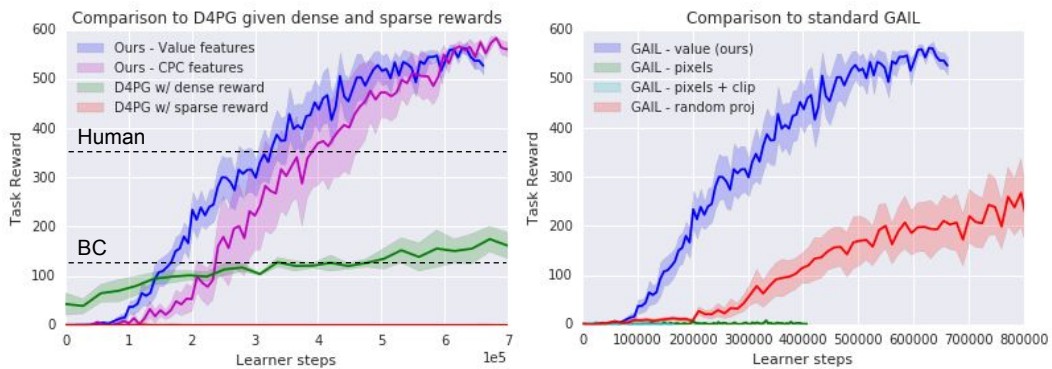

Figure 4: Comparison of our methods with D4PG baselines in dense and sparse reward scenarios.

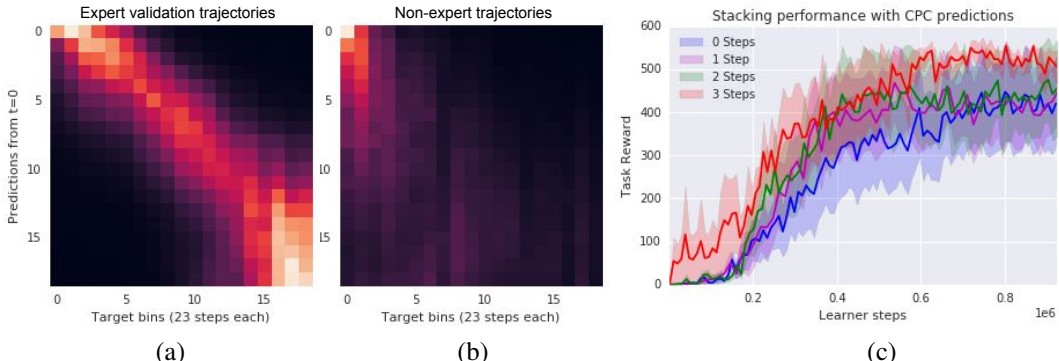

Figure 5: Visualization of CPC predictions on (a) expert sequences and (b) non-expert trajectories. The $i^{th}$ row and $j^{th}$ column in the matrices display the CPC's probability of the $j^{th}$ frame being $j$ steps away from the $i^{th}$ frame. Visualizations are averaged over all the expert and non-expert trajectories. Note that CPC models the future observations well for expert sequences, but not non-expert. (c) shows that conditioning on k-step predictions improves the performance of our method on stacking tasks, when the discriminator also uses CPC embeddings.

In figure 4, we see that D4PG with sparse rewards never takes off due to the complexity of exploration in this task, whereas with dense rewards its learning pace is very slow. However our imitation methods learn very quickly with superior performance despite the fact that they only utilize sparse rewards. Using our method, the agent using value network features takes off more quickly than with CPC features, though they reach to a comparable performance towards the end. The conventional GAIL from pixels perform very poorly, and GAIL with tiny adversaries on random projections achieves limited success. Note that with CPC features, which are of dimension 128, the discriminator network has only 128 parameters. The value network features are 2048-dimensional.

One possible reason that GAIL value features worked while pixel features did not could be due to the regularizing effect of norm clipping applied in the critic optimizer. To check for this, we evaluated another agent indicated by "GAIL - pixels + clip" which performs norm clipping via `tf.clip_by_global_norm(..., 40)}`as is done in the critic. We find that as in the pixel case, this does not result in success in either Jaco (figure 4) or Walker2D (figure 7a).

In addition to using CPC features as input to the discriminator, we can also try to make use of the temporal predictions made by CPC. At each step, one can query CPC about what the expert state would look like several steps in the future, starting from the current state. Figure 5 visualizes CPC features learned from the 500 training trajectories on the held out validation trajectories.

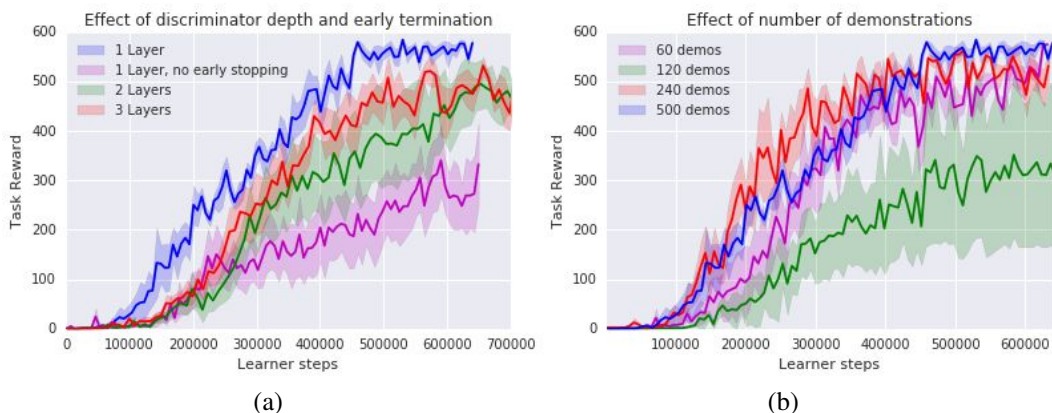

(a)                        (b)

Figure 6: Jaco stacking ablation experiments. Left: We find that adding layers to the discriminator network does not improve performance, and not doing early termination hurts performance. Right: In the case of 120 demonstrations one of the three seeds failed to take off. However, even with 60 demonstrations, our agent can learn stacking equivalently well as with 500.

## 4.2 ABLATION EXPERIMENTS

In the previous Jaco arm experiments we showed that learning a discriminator directly on the pixels resulted in poor performance. In the first ablation experiment we determine if a tiny (linear) discriminator is the optimal choice for imitation learning, or if a deeper network on these features can improve the results. Figure 6(a) shows the effect of the number of layers: as the discriminator becomes more powerful the performance actually degrades, indicating the advantage of a small discriminator on a meaningful representation.

In Section 3 we introduced an early termination criterion to stop an episode when the discriminator score becomes too low. Figure 6(a) shows that that when early stopping is disabled, the model learns a lot slower. To understand better why this helps learning we plot the average episode length during training in Figure 7. In the beginning of training the discriminator is not good at distinguishing expert trajectories from agent trajectories yet which is why the episode length is high. After a while (2000+ episodes) most of the trajectories get stopped early on in the episode. From 6000 episodes onwards the agent becomes better at imitating the expert and the episodes take longer. The same figure shows the task and imitation reward (which are scaled between 0 and 1).

In a third ablation experiment we evaluate the data efficiency of the proposed method in terms of expert demonstrations. Figure 6(b) visualizes the performance with 60, 120, 240 and 500 demonstrations, showing that even 60 demonstrations is enough to get good performance. We believe the result with 120 demos is an outlier: one of the three random seeds did not take off.

## 4.3 WALKER2D EXPERIMENTS

Figure 7 a shows results on the planar walker. As in the Jaco experiments, the conventional GAIL on pixels with and without norm clipping did not learn. Both our proposed method using value network features and using random projections learn to run. Videos of the trained agent are included in the supplementary videos anonymously linked in the appendix.

## 4.4 IMITATION LEARNING WITHOUT REWARDS

In this section we show results for agents trained without any rewards, as described in section 3.3. We used expert states in the final $1/3$rd of each sequence as positive examples for $D_{goal}$, and set $\alpha = 0.5$ in the imitation reward function.

Figure 8 (left) shows different runs (each with a different random seeds), showing that two out of five runs were able to learn without providing any task reward. The best no-task-reward agent seed achieved a success rate of 55%.

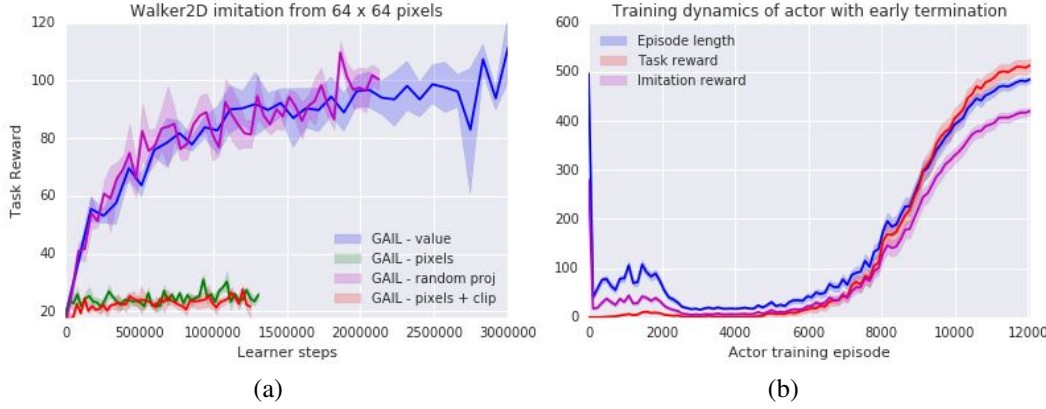

Figure 7: (a): Walker2D experiments. Both random projections and value net features yield agents that learn to run, while pixel features even with norm clipping do not. (b): Training dynamics on Jaco block stacking with early termination.

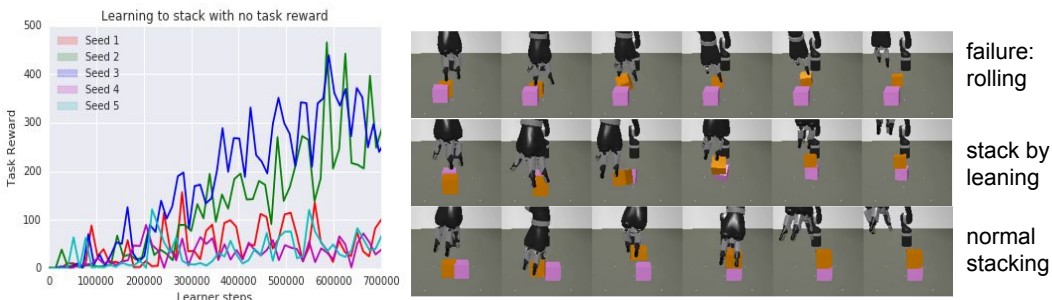

Figure 8: Example agent trajectories when trained only with imitation rewards. In the top row, the agent learns to stack in a more efficient way than the demonstrator, taking under 2s while the human teleoperator takes up to 30s. In the bottom row, we see an agent exploit, in which the top block is rolled to the background to give the appearance of a completed stack, without actually stacking.

## 5 RELATED WORK

The idea of leveraging expert demonstrations to improve agent performance has a long history in robotics (Bakker & Kuniyoshi, 1996; Kawato et al., 1994; Miyamoto et al., 1996). Similar in spirit to very recent work, Schaal (1997) show that by priming a Q-function on expert demonstrations, their Q-learning agent can perform cart-pole swing up after a single episode of training. However, our task is different in that we do not assume access to both states and actions, only pixel observations, so we cannot prime the value function on expert demonstrations in the same way.

A large amount of work in the past several years has tried to extend the success of deep learning beyond discriminative tasks in computer vision, towards taking actions to interact with simulated or real environments. Imitation learning is an attractive setting from this perspective, because the inputs and outputs to the learning problem strongly resemble those found in classification and regression problems at which deep networks already excel in solving.

A simple yet effective approach is supervised imitation (Pomerleau, 1989), also called behavioral cloning. Duan et al. (2017) extend this to one-shot imitation, in which a behaviors are inferred from single demonstrations via an encoder network, and a state-to-action decoder with an attention mechanism replicates the desired behavior on a new problem instance. This approach is able stack blocks into target arrangements from scripted demonstrations on a simulated Fetch robot. Instead of using attention, (Finn et al., 2017) use a gradient-based meta learning approach to perform one-shot learning of observed behaviors. This approach learns to pick and place novel objects into containers given video demonstrations on a PR2 robot. Our approach is different mainly in that we aim for the agent to learn by interacting with the environment rather than supervised learning.

A major downside of behavioral cloning is that, if the agent ventures into very different states than were observed in the expert trajectories, this results in cascading failures. This necessitates a large number of demonstrations and limits how far the agent can generalize. It also requires access to demonstrator actions, which may not always be available, and tend to be tightly coupled to the particular robot and teloperation setup used to gather demonstrations.

Instead of using behavior cloning, Ziebart et al. (2008); Ng et al. (2000); Abbeel & Ng (2004) propose inverse reinforcement learning (IRL), in which they learn a reward function from demonstrations, and then use reinforcement learning to optimize that learned reward. Hester et al. (2018) developed deep Q-Learning from demonstration (DQfD), in which expert trajectories are added to experience replay and jointly used to train agents along with their own experiences. This was later extended by Pohlen et al. (2018) to better handle sparse-exploration Atari games. Vecerik et al. (2017) develop deterministic policy gradients from demonstration (DPGfD), and similarly populate a replay buffer with both expert data and agent experience, and show that through imitation it can solve a peg insertion task on a real robot without access to any dense shaped reward. However, both of these methods still require access to the expert actions in order to learn.

Following the success of Generative Adversarial Networks Goodfellow et al. (2014) in image generation, GAIL (Ho & Ermon, 2016) applies adversarial learning to the problem of imitation. Although many variants are introduced in the literature (Li et al., 2017; Fu et al., 2018; Zhu et al., 2018; Baram et al., 2017), making GAIL work for high-dimensional input spaces, particularly for hard exploration problems with sparse rewards, remains a challenge. Our major contribution, which is complementary, is that through the use of minimal adversaries on top of learned features we can successfully solve sparse reward tasks with high-dimensional input spaces.

Another line of work is learning compact representations for imitation learning using expert observations (i.e. no actions). Both (Sermanet et al., 2018) and (Aytar et al., 2018) learn self-supervised features from third person observations in order to mitigate the domain gap between first and third person views. At the end, they both utilize these feature spaces for closely tracking a single expert trajectory, whereas we use our features for learning the task using all available expert trajectories through GAIL. Our target is not to track a single trajectory, but to generalize all possible initializations of a hard exploration task. We utilize both static self-supervised features such as contrastive predictive coding (van den Oord et al., 2018), and dynamic value network features which constantly change during the learning process, and show that both can be used to successfully train block stacking agents from sparse rewards on pixels.

## 6 CONCLUSIONS

In this work, we developed a very simple off-policy GAIL agent that solves a challenging sparse reward task of block stacking from video demonstrations. The proposed agent does not require demonstrator actions or proprioceptive states, or any form of special initialization, tracking or warm starts. Surprisingly, the discriminator network can consist of a single layer, as few as 128 parameters, which may force it to focus only on distinguishing high-level differences between agent and expert. Finally, without any task reward at all, we trained agents that stack faster than the human demonstrators, showing that the performance of adversarial imitation is not necessarily bounded by the performance of the human demonstrator.

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

## 7 APPENDIX

### 7.1 SUPPLEMENTARY VIDEOS

Videos of our learned agents can be viewed at the following anonymized web site:
https://sites.google.com/view/iclr2019-visual-imitation/home

### 7.2 BEHAVIOR CLONING BASELINE MODEL

In figure 4 we include a dashed line labeled "BC" indicating the average performance of a pure supervised baseline model that regresses expert actions from pixel observations.

The behavior cloning model consists of the same residual network pixel encoder architecture that we use for D4PG, followed by a 128-dimension LSTM, followed by a final 512-dimensional linear layer, ELU, then the output actions. The stacking accuracy is approximately 15%.

### 7.3 CONTRASTIVE PREDICTIVE CODING MODEL

Figure 9 (left) shows a visualization of CPC on video data. The model consists of two parts: the encoder which maps every observation $x_t$ to a latent representation $z_t = g_{\text{enc}}(x_t)$ (or target vector) and the autoregressive model which summarizes the past latents into a context vector $c_t = g_{\text{ar}}(z_{\leq t})$. Both optimize the same loss:

$$\mathcal{L}_{\text{CPC}} = - \underset{X}{\mathbb{E}} \left[ \log \frac{\exp(z_{t+k}^T W_k c_t)}{\sum_{j=1}^{N} \exp(z_j'^T W_k c_t)} \right],$$

where $z_1', z_2' \ldots z_N'$ are negative samples, which in the case of CPC are usually drawn from other examples or timesteps in the minibatch. The weights $W_k$ for the bilinear mapping $z_{t+k} W_k c_t$ are also learned and depend on $k$, the number of latent steps the model is predicting in the future.

By optimizing $\mathcal{L}_{\text{CPC}}$ the mutual information between $z_{t+k}$ and $c_t$ is maximized, which results in the variables that the context and target have in common being linearly embedded into compact representations. This is especially useful for extracting slow features, for example when $z_{t+k}$ and $c_t$ are far apart in time.

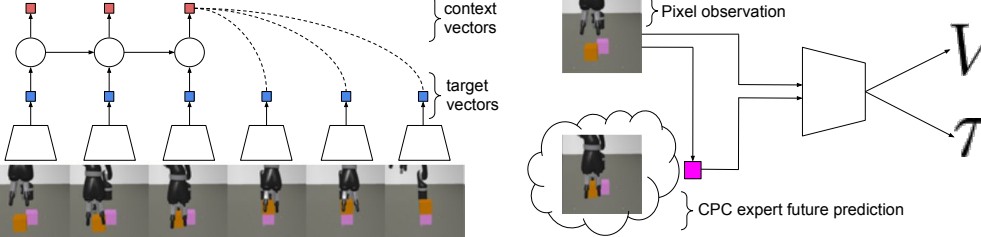

Figure 9: Overview of our proposed approach. Left: model learning via contrastive predictive coding (CPC). Right: After training and freezing CPC expert model, we train the agent, which makes use of CPC future predictions. Note that we never need to predict in pixel space.

### 7.4 TASK REWARD FUNCTIONS

Our reward functions are slightly modified from Zhu et al. (2018). For evaluation each episode lasts 500 time steps; we do not use early stopping.

**Dense staged reward**: We define five stages and their rewards to be initial (0), reaching the orange block (0.125), lifting the orange block (0.25), stacking the orange block onto the pink block (1.0), and releasing the orange block and lifting the arm (1.25).

**Sparse reward**: We define two stages and their rewards to be initial (0) and stacking the orange block onto the pink block (1.0). There are no rewards for reaching, lifting or releasing.

## 7.5 HYPERPARAMETERS

Actor and critic share a residual network with twenty convolutional layers (3x3 convolutions, four 5-layer blocks with 64, 128, 256 and 512 channels), instance normalization (Ulyanov et al.) and exponential linear units (Clevert et al., 2015) between layers and a fully connected layer with layer normalization (Ba et al., 2016). Both actor and critic use independent three-layer fully connected networks (2x1024 and an output layer) with exponential linear unit between layers.

| Parameters | Values |
|---|---|
| Input Width | 128 |
| Input Height | 128 |
| **D4PG Parameters** | |
| $V_{min}$ | 0 |
| $V_{max}$ | 100 |
| $V_{bins}$ | 101 |
| N step | 5 |
| Actor learning rate | $10^{-4}$ |
| Critic learning rate | $10^{-4}$ |
| Optimizer | Adam (Kingma & Ba (2014)) |
| Batch size | 256 |
| Target update period | 100 |
| Discount factor ($\gamma$) | 0.99 |
| Replay capacity | $10^6$ |
| Number of actors | 128 |
| **Imitation Parameters** | |
| Discriminator learning rate | $10^{-3}$ |

## 7.6 D4PG AGENT DESCRIPTION

Instead of using a scalar state-action value function, we adopt Distributional $Q$ fuctions where $Q(s, a|\theta) = \mathbb{E}Z(s, a|\theta)$ for some random variable $Z$. In this paper, we adopt a categorical representation of $Z$ such that $Z(s, a|\theta) = z_i$ w.p. $p_i \propto \exp(\omega_i(s, a|\theta))$ for $i \in \{0, \cdots, l-1\}$. The $z_i$'s are fixed atoms bounded between $V_{\min}$ and $V_{\max}$ such that $z_i = V_{\min} + i\frac{V_{\max} - V_{\min}}{l-1}$.

Again following (Barth-Maron et al., 2018), we compute our bootstrap target with N-step returns. Given sub-sequence $s_t, a_t, \{r_t, \cdots, r_{t+N-1}\}, s_{t+N}$, we construct a bootstrap target $Z'$ such that $Z' = z'_i$ w.p. $p'_i \propto \exp(\omega_i(s_{t+K}, \pi(s_{t+K}|\phi')|\theta'))$ where $z'_i = \sum_{n=0}^{N-1} \gamma^j r_{t+n} + \gamma^N z_i$. Notice $Z'$ is not likely have the same support as $Z$. We therefore adopt the categorical projection $\Phi$ proposed in Bellemare et al. (2017). The loss function for training our distributional value functions is

$$\mathcal{L}_N(\theta) = \mathbb{E}_{(s_t, a_t, \{r_t, \cdots, r_{t+N-1}\}, s_{t+N}) \sim \mathcal{B}} \left[ H(\Phi(Z'), Z(s_t, a_t|\theta)) \right], \qquad (7)$$

where $H$ represents cross entropy. Finally, we use distributed prioritized experience replay (Horgan et al., 2018; Schaul et al., 2016) to further increase stability and learning efficiency.

