# OpenReview forum: "Visual Imitation with a Minimal Adversary"
_ICLR.cc/2019/Conference_

### Official Review · AnonReviewer2 · 2018-11-02
**Potentially practical improvement of sparse-reward RL using IL, but a bit unclear when it helps**

**Rating:** 6
**Confidence:** 4

**Review:**

The submission describes a sort of hybrid between reinforcement learning and imitation learning, where an auxiliary imitation learning objective helps to guide the RL policy given expert demonstrations.  The method consists of concurrently maximizing an RL objective--augmented with the GAIL discriminator as a reward—and minimizing the GAIL objective, which optimizes the discriminator between expert and policy-generated states.  Only expert states (not actions) are required, which allows the method to work given only videos of the expert demonstrations.  Experiments show that adding the visual imitation learning component allows RL to work with sparse rewards for complex tasks, in situations where RL without the imitation learning component fails.

Pros:
+ It is an interesting result that adding a weak visual imitation loss dramatically improves RL with sparse rewards
+ The idea of a visual imitation signal is well-motivated and could be used to solve practical problems
+ The method enables an ‘early termination’ heuristic based on the imitation loss, which seems like a nice heuristic to speed up RL in practice

Cons:
+ It seems possible that imitation only helps RL where imitation alone works pretty well already
+ Some contributions are a bit muddled: e.g., the “learning with no task reward” section is a little confusing, because it seems to describe what is essentially a variant of normal GAIL
+ The presentation borders on hand-wavy at parts and may benefit from a clean, formal description

The submission tackles a real, well-motivated problem that would appeal to many in the ICLR community.  The setting is attractive because expert demonstrations are available for many problems, so it seems obvious that they should be leveraged to solve RL problems—especially the hardest problems, which feature very sparse reward signals.  It is an interesting observation that an imitation loss can be used as  a dense reward signal to supplement the sparse RL reward.  The experimental results also seem very promising, as the imitation loss seems to mean the difference between sparse-reward RL completely failing and succeeding.  Some architectural / feature selection details developed here seem to also be a meaningful contribution, as these factors also seem to determine the success or failure of the method.

My biggest doubt about the method is whether it really only works where imitation learning works pretty well already.  If we don’t have enough expert examples for imitation learning to work, or if the expert is not optimizing the given reward function, then it is possible that adding the imitation loss is detrimental, because it induces an undesirable bias.  If, on the other hand, we do have enough training examples for imitation learning to succeed and the expert is optimizing the given reward function, then perhaps we should just do imitation learning instead of RL.  So, it is possible that there is some sweet spot where this method makes sense, but the extent of that sweet spot is unclear to me.

The experiments are unclear on this issue for a few reasons.  First, figure 4 is confusing, as it is titled ‘comparison to standard GAIL', which makes it sound like a comparison to standard imitation learning.  However, I believe this figure is actually showing the performance of different variants of GAIL used as a subroutine in the hybrid RL-IL method.  I would like to know how much reward vanilla GAIL (without sparse rewards) achieves in this setting.  Second, figure 8 seems to confirm that some variant of vanilla imitation learning (without sparse rewards) actually does work most of the time, achieving results that are as good as some variants of the hybrid RL-IL method.  I think it would be useful to know, essentially, how much gain the hybrid method achieves over vanilla IL in different situations.

Another disappointing aspect of the paper is the ‘learning with no task reward’ section, which is a bit confusing.  The concept seems reasonable at a first glance, except that once we replace the sparse task reward with another discriminator, aren’t we firmly back in the imitation learning setting again?  So, the motivation for this section just seems a bit unclear to me.  This seems to be describing a variant of GAIL with D4PG for the outer optimization instead of TRPO, which seems like a tangent from the main idea of the paper.  I don’t think it is necessarily a bad idea to have another discriminator for the goal, but this part seems somewhat out of place.

On presentation: I think the presentation is a bit overly hand-wavy in parts.  I think the manuscript could benefit from having a concise, formal description.  Currently, the paper feels like a series of disjoint equations with unclear connections among them.  The paper is still intelligible, but not without knowing a lot of context relating to RL/IL methods that are trendy right now.  I feel that this is an unfortunate trend recently that should be corrected.  Also, I’m not sure it is really necessary to invoke “GAIL” to describe the IL component, since the discriminator is in fact linear, and the entropy component is dropped.  I think “apprenticeship learning” may be a more apt analogy.

On originality: as far as I can tell, the main idea of the work is novel.  The work consists mainly of combining existing methods (D4PG, GAIL) in a novel way.  However, some minor novel variations of GAIL are also proposed, as well as novel architectural considerations.

Overall, this is a nice idea applied to a well-motivated problem with promising results, although the exact regime in which the method succeeds could be better characterized.

---

> ### Author Response · Authors · 2018-11-19
> **Imitation to help RL with task rewards, and better RL imitation in general**
>
> We thank AR2 for your thorough feedback.
>
> AR2 writes that “imitation may only help RL when imitation alone works well already”. First we should point out that imitation can also itself be RL, as in the case of our model where a reward function is derived from the discriminator score.
>
> For example in walker2D, none of our experiments use human specified task reward functions, but the agents learn from experience using RL on the discriminator score as a reward function. Rather than “imitation helping RL”, the contribution here is simply “better RL imitation”.
>
> In the case where we use human task rewards - Jaco stacking - we show that by using imitation, we can replace dense staged task rewards with sparse task rewards, which is a big improvement - a clear case of “imitation helping RL”.  Although figure 8 shows that we can sometimes learn to stack without human crafted sparse rewards, we were never able to learn stacking agents with “reward vanilla GAIL”.  We hope this is sufficient to address AR2’s first point in Cons, and we will clarify this point in the paper as well.
>
> AR2 points out that the “learning with no task rewards” section is (1) muddled and is (2) essentially a variant of normal GANS. As to the first point, we will try to clarify the presentation (perhaps adding pseudocode to better describe exactly what we are doing?). For the second point, we agree - it is precisely an auxiliary discriminator network but otherwise a normal vanilla GAN. However, it does something quite useful - replacing a previously hand-engineered reward function that required access to block and arm positions! The fact that such a simple GAN setup can be arranged to do this from pixels should be great news to practitioners and perhaps place this point in the Pros section instead of the Cons.
>
> Hand-wavy presentation: we agree that the presentation could use more precision and clarity. We tried to emphasize the simplicity of our adversarial setup, but may have erred on the side of too few details. We will try to improve overall clarity in the final version.

---

> > ### Comment · AnonReviewer2 · 2018-12-04
> > **RL as a problem vs RL as an algorithm**
> >
> > I just wanted to clarify one important point.  I want to make a distinction between RL as a problem and RL as an algorithm.  When I think of RL as a problem, I think of a situation where I want to maximize some task reward that is meaningful unto itself.  We solve these problems, obviously, with RL algorithms.  However, we can also apply RL algorithms to problems with RL substructure, where the reward is some kind of intermediate quantity.  Imitation learning with GAIL would fall into the latter category: we are incidentally applying an RL algorithm to solve a supervised learning problem.
> >
> > To achieve a given task, I might be able to formulate the problem either as RL or supervised/imitation learning.  If it is easy to manually specify a reward function to achieve the task, then RL is almost always preferable because it requires no training data.  Unfortunately, since RL is hard, one might in practice resort to applying IL even if we can specify a reward function.
> >
> > My point was that for the method to be practically useful, it needs to solve problems where both RL and IL were previously intractable.  Let's say RL fails for task A, but I have a budget of N training examples to generate.  If I train vanilla IL using those N examples, and the resulting policy solves the task, then I might as well do vanilla IL.  On the other hand, if vanilla IL fails in this case, but method B somehow combines RL and IL to solve the task, then method B is practically useful.

---

> > > ### Author Response · Authors · 2018-12-09
> > > **Imitation vs supervised learning, Usefulness**
> > >
> > > Imitation can be treated as a supervised learning problem when there are (state, action) pairs available and you want to learn a policy by regressing expert actions from the states. However, when no expert actions are available to predict, one must learn from experience by interacting with the environment. This type of imitation therefore becomes an RL problem, not a supervised learning problem. If you want to distinguish it from RL on human-specified, static reward functions, it might be useful to call this "RL imitation".
> > >
> > > This is the case for GAIL using only expert states (no actions), and in our current work. It is not the case that we are incidentally applying an RL algorithm to solve a supervised learning problem. Without expert actions, there is no way to formulate the problem as supervised learning. Imitation learning is not equivalent to supervised learning, although supervised learning can be used for imitation in some, but not all cases.
> > >
> > > By your usefulness criterion, our proposed model has shown itself to be useful. Supervised learning was intractable because we assume no expert actions were provided, and RL on sparse human-specified task rewards for stacking failed to learn the task. In your proposed taxonomy this corresponds to "both RL and IL were previously intractable". Our proposed RL imitation model solved that task with high success rate. Even when we compared to a (supervised) IL baseline with additional information of expert actions, our proposed approach worked much better.

---

### Official Review · AnonReviewer3 · 2018-11-04
**straightforward idea, but this approach may not be applicable in general applications**

**Rating:** 3
**Confidence:** 3

**Review:**

This paper aims at solving the problem of estimating sparse rewards in a high-dimensional input setting. The authors provide a simplified version by learning the states from demonstrations. This idea is simple and straightforward, but the evaluation is not convincing.

I am wondering if this approach still works in more general applications, e.g., when state distributions vary dramatically or visual perturbations arise in the evaluation phase.

In addition, it is weird to use adversary scheme to estimate rewards. Namely, the agent is trying to maximize the rewards, but the discriminator is improved so as to reduce rewards.

In section 3, the authors mention an early termination of the episode, this is quite strange in real applications, because even the discriminator score is low the robot still needs to accomplish the task.

Finally, robots are subject to certain physical constraints, this issue can not be addressed by merely learning demonstrated states.

---

> ### Author Response · Authors · 2018-11-19
> **Please review the literature on adversarial imitation (GAIL)**
>
> We thank AR3 very much for providing feedback. However, we think AR3 has missed several of the key points of our paper, which we will try to clarify below and in the final version of our paper as needed.
>
> First, our goal is not to estimate sparse rewards, but to train agents to solve continuous control tasks from pixel observations using raw video demonstrations, without access to proprioceptive states. There may be sparse rewards or no rewards available, aside from imitation-based rewards.
>
> AR3 also suggests that it is weird to use an adversary scheme to estimate rewards. However, this is actually a well established and effective approach; see for example
>     Ho, Jonathan, and Stefano Ermon. "Generative adversarial imitation learning." Advances in Neural Information Processing Systems. 2016.
> which currently has over 200 citations. What we contribute in this paper is showing how to extend this method to learning robot manipulation policies from raw video.
>
> AR3 is basically correct in pointing out that “the agent is trying to maximize rewards, but the discriminator is improved so as to reduce rewards”. This is a fundamental tension inherent in any adversarial learning setup, not a flaw particular to our approach.
>
> AR3 is justifiably concerned with the proposed early termination scheme, since ultimately we want the robot to attempt to finish the task regardless of the discriminator score. During evaluation / test time, this is true, which is why we only apply early termination during training. We will update the paper to clarify about this.
>
> AR3 wonders whether this approach could work in more general settings, e.g. where state distributions vary dramatically. There is early work in this direction for visually much simpler domains (see e.g. “Third person imitation learning” in ICLR’17) and in visual domains with behavior cloning agents. However, learning from visual experience on a robot from dramatically varying third person observations, remains a grand challenge for the field. We agree with AR3 that it is a worthy goal, but also not in scope for this paper.

---

### Official Review · AnonReviewer1 · 2018-11-11
**Sample complexity experiments are interesting, but the ideas presented seems to overlap ideas from existing work.**

**Rating:** 5
**Confidence:** 4

**Review:**

---
Update: I think the experiments are interesting and worthy of publication, but the exposition could be significantly improved. For example:

- Not sure if Figure 1 is needed given the context.
- Ablation study over the proposed method without sparse reward and hyperarameter \alpha
- Move section 7.3 into the main text and maybe cut some in the introduction
- More detailed comparison with closely related work (second to last paragraph in related work section), and maybe reduce exposition on behavior cloning.

I like the work, but I would keep the score as is.
---


The paper proposes to use a "minimal adversary" in generative adversarial imitation learning under high-dimensional visual spaces. While the experiments are interesting, and some parts of the method has not been proposed (using CPC features / random projection features etc.), I fear that some of the contributions presented in the paper have appeared in recent literature, such as InfoGAIL (Li et al.).

- Use of image features to facilitate training: InfoGAIL used pretrained ResNet features to deal with high-dimensional inputs, only training a small neural network at the end.
- Tracking and warm restarts: InfoGAIL does not seem to require tracking a single expert trajectory, since it only classifies (s, a) pairs and is agnostic to the sequence.
- Reward augmentation: also used in InfoGAIL, although they did not use sparse rewards for augmentation.

Another contribution claimed by this paper is that we could do GAIL without action information. Since we can shape the rewards for most of our environments that do not depend on actions, it is unsurprising that this could work when D only takes in state information. However, it is interesting that behavior cloning pretraining is not required in the high-dimensional cases; I am interested to see a comparison between with or w/o behavior cloning in terms of sample complexity.

One setting that could potentially be useful is where the expert and policy learner do not operate within the same environment dynamics (so actions could not be same) but we would still want to imitate the behavior visually (same state space).

The paper could also benefit from clearer descriptions, such as pointers to which part of the paper discusses "special initialization, tracking, or warm starting", etc., from the introduction.

---

> ### Author Response · Authors · 2018-11-19
> **Key differences from previous work; comparison to behavior cloning**
>
> Thanks to AR1 for your detailed comments and pointing out relevant previous work. Below we address each part of the feedback.
>
> We agree that InfoGAIL shares significant motivation with our model in that it learns from pixels.
>
> However, our work makes several advances that will be of interest to the research community:
> - InfoGAIL was demonstrated in the TORCS driving game with discrete actions, whereas our model is applied to challenging continuous control tasks such as block stacking.
> - While our method can be used with pre-trained features as in InfoGAIL, our best performing method uses deep value network features, which are trained together with the discriminator reward function, so no feature pre-training is needed in our model.
> - InfoGAIL used (state, action) pairs, whereas we do not use any expert actions.
> - We show that discriminator-based early stopping can improve sample complexity.
> - We show that it is possible to replace human engineered rewards with auxiliary discriminators on the Jaco block stacking task.
>
> Furthermore, our approach could easily be combined with that of InfoGAIL. Our goal in the paper was to show that with our approach, even the most naive GAN could be used to solve challenging visual imitation tasks. Using the latest adversarial learning techniques - e.g. information theoretic objective as in InfoGAIL - could improve things further.
>
> Comparison to behavior cloning, sample efficiency:
> - In terms of demonstration efficiency, we can perform much better than behavior cloning with a small fraction of the demonstrations.
> - With only 60 demonstrations, we can achieve >90% stacking success rate, whereas a comparable behavior cloning agent with 500 demonstrations only achieved ~33% success rate.
>
> Application to third-person imitation (expert and agent may have differing dynamics):
> - This is a great suggestion, probably beyond the scope of our current paper. However, the fact that expert actions are not needed in our approach potentially removes an important barrier to this line of research.
>
> Clarity:
> - We agree that the descriptions and terminology can be improved, which will be forthcoming in the final version of the paper.

---

### Meta-Review · Area_Chair1 · 2018-12-16
**imitation in with high dimensional observations - contributions not sufficiently validated in experiments**

**Confidence:** 5
**Recommendation:** Reject

**Metareview:**

The paper extends an existing approach to imitation learning, GAIL (Generative Adversarial Imitation Learning, based on an adversarial approach where a policy learner competes with a discriminator) in several ways and demonstrates that the resulting approach can learn in settings with high dimensional observation spaces, even with a very low dimensional discriminator. Empirical results show promising performance on a (simulated) robotics block stacking task, as well as a standard benchmark - Walker2D (DeepMind control suite).

The reviewers and the AC note several potential weaknesses. Most importantly, the contributions of the paper are "muddled" (R2). The authors introduce several modifications to their baseline, GAIL, and show empirical improvements over the baseline. However, the presented experiments do systematically identify which modifications have what impact on the empirical results. For example, R2 mentions this for figure 4, where it appears on first look that the proposed approach is compared to the vanilla GAIL baseline - however, there appear to be differences from vanilla GAIL, e.g., in terms of reward structure (and possibly other modeling choices - how close is the GAIL implementation used to the original method, e.g., in terms of the policy learner and discriminator)? There is also confusion on which setting is addressed in which part of the paper, given that there is both a "RL+IL" and an "imitation only" component.

In their rebuttal, the authors respond to, and clarify some of the questions raised by the reviewers, but the AC and corresponding reviewers consider many issues to remain unclear. Overall, the presentation could be much improved by indicating, for each set of experiments, what research question or hypothesis it is designed to address, and to clearly indicate conclusions on each question once the results have been discussed. In its current state, the paper reads as a list of interesting and potentially highly valuable ideas, together with a list of empirical results. The real value of the paper should come in when these are synthesized into lessons learned, e.g., why specific results are observed and what novel insights they afford the reader. Overall, the paper will benefit from a thorough revision and is not considered ready for publication at ICLR at this stage.

The AC notes that they placed less weight on R3's assessment, due to their relatively low confidence, because they appear not to be familiar with key related work (GAIL), and did not respond to further requests for comments in the discussion phase.

The AC also notes a potential weakness that was not brought up by the reviewers, and which they therefore did not weigh into their assessment of the paper, but nevertheless want to share to hopefully help improve a future version of the paper. Figure 6(b) should be interpreted with caution given that performance with a greater number of demonstrations (120 vs 60) showed lower performance. The authors note in the caption that one of the "120 demos" runs "failed to take of". This suggests that variance for all these runs may be underestimated with the currently used number of seeds. It is not clear what the shaded region indicates (another drawback) but if I interpret these as standard errors then this plot would suggest lower performance for higher numbers of demonstrations with some confidence - clearly that conclusion is unlikely to be correct.